# Chromium Transport and Fate in Vadose Zone: Effects of Simulated Acid Rain and Colloidal Types

**DOI:** 10.3390/ijerph192416414

**Published:** 2022-12-07

**Authors:** Wenjing Zhang, Kaichao Zhao, Bo Wan, Zhentian Liang, Wenyan Xu, Jingqiao Li

**Affiliations:** 1Key Laboratory of Groundwater Resources and Environment, Ministry of Education, Jilin University, Changchun 130021, China; 2College of New Energy and Environment, Jilin University, Changchun 130021, China; 3Chemical Geological Prospecting Institute of Liaoning Province Co., Ltd., Jinzhou 121007, China; 4Songliao Water Resources Commission, Ministry of Water Resources, Changchun 130021, China

**Keywords:** acid rain, colloid, chromium, release, vadose zone

## Abstract

Chromium (Cr) can enter groundwater through rainfall infiltration and significantly affects human health. However, the mechanisms by which soil colloids affect chromium transport are not well investigated. In this study, column experiments were conducted to simulate the chromium (Cr) transport mechanism in two typical soils (humic acid + cinnamon soil and montmorillonite + silt) in the vadose zone of a contaminated site and the effects of acid rain infiltration conditions. The results showed that Mt colloids have less influence on Cr. The fixation of Cr by colloid mainly occurs in the cinnamon soil layer containing HA colloid. The adsorption efficiency of Cr was increased by 12.8% with the addition of HA. In the HA-Cr system, the introduction of SO4^2−^ inhibited the adsorption of Cr, reducing the adsorption efficiency from 31.4% to 24.4%. The addition of Mt reduced the adsorption efficiency of Cr by 15%. In the Mt-Cr system, the introduction of SO4^2−^ had a promoting effect on Cr adsorption, with the adsorption efficiency increasing from 4.4% to 5.1%. Cr release was inhibited by 63.88% when HA colloid was present, but the inhibition owing to changes in acidity was only 14.47%. Mt colloid promotes Cr transport and increases the leaching rate by 2.64% compared to the absence of Mt. However, the effect of acidity change was not significant. Intermittent acid rain will pose a higher risk of pollutant release. Among the influencing factors, the type of colloid had the most significant influence on the efficiency of Cr leaching. This study guides the quantitative assessment of groundwater pollution risk caused by Cr in the vadose zone.

## 1. Introduction

The most accessible source of fresh water is groundwater, which is essential for industry, agriculture, and humans [1,2,3,4]. However, the different microbial and chemical contamination are frequently challenges [5]. Chromium (Cr) enters the groundwater through other sources, such as rainfall infiltration, which significantly affects human health. It is widely used in various industrial activities. Chromium is often present in the soil in the trivalent and hexavalent oxidation states. The toxicity of chromium is directly related to its valence state, with hexavalent chromium being much more toxic than trivalent chromium. Cr (VI) has a strong migration ability and can rapidly migrate [4,5]. It is a highly toxic pollutant that can cause plant growth restriction, animal chromosomal mutations, human allergies, mucosal atrophy, and cancer [6]. Cr (VI) has a structure that is parallel to SO_4_^2−^ and possesses high biological availability [7]; thus, it can cause considerable damage to organisms.

The vadose zone is a complex system containing solids, liquids, and gases. Most of the pollutions are intercepted, but some still enter the groundwater [8]. The soil structure and properties determine how the vadose zone will block pollutants [9]. Massive chromium contamination incidents have been reported in South Africa, New Jersey, and Maryland, with total chromium levels reaching 61,000 mg/kg [10,11,12,13]. China also suffers from soil Cr contamination, and approximately 1.1% of soil throughout the country has a Cr content above the environmental limit of 30 mg/kg [14]. In Qujing, Yunnan, China, the soil concentration of Cr in the vadose zone has reached 3384 mg/kg. Contamination in the vadose zone can directly or indirectly make contact with organisms through surface runoff, soil flow, or wind; this contact can harm the organism. Pollutants in the vadose zone can also migrate vertically into the groundwater and spread through hydraulic action [15].

Chromium is pathogenic to humans and susceptible to colloidal influences in groundwater. An increasing number of studies have focused on groundwater pollution caused by the release of hexavalent chromium from the soil, for example, investigating the release of Cr owing to mineral weathering under alkaline conditions and assessing the risk of this release [16]. The risk of Cr release in riverbank areas where soil and groundwater interact indicates that acidity affects Cr release [17]. Attempts have been made to reduce the risk of groundwater pollution by reducing Cr release through solidification and stabilization techniques [18,19]. Alternatively, reduction technology can be used to transform Cr into a lower valence state to enhance its fixation and reduce the risk of its release [20]. However, the technology used to convert Cr (VI) to Cr (III) still needs to overcome the problem of “yellow return”, where the Cr (VI) concentration in the reduced chromium slag appears to bounce back up with stockpile time [21]. Studies have failed to elucidate potential pollution sources, explain why acidic conditions can increase the risk of Cr diffusion [14], or quantitatively assess the risk of groundwater pollution. These failures are mainly related to the complex structure and material composition of the vadose zone.

Humic acid and montmorillonite colloids widely present in the vadose zone have a large specific surface area and many functional groups; they can limit the migration of chromium to a certain extent and cause heavy metals to accumulate in the vadose zone [22,23,24]. A large amount of colloid in the vadose zone of the study area trapped a considerable amount of Cr, forming a potential pollution source. In aquifers, acid rain can promote the release of Cr bound to iron and manganese oxides, but the mechanism for this is unknown [18,25]. Acid rain can also increase the release of carbonate-bound, exchangeable, and organic-bound Cr, mainly by promoting the morphological transformation of Cr [26]. Therefore, colloids and acid rain may be key factors affecting Cr release in the vadose zone.

Currently, most studies have focused on the redox of chromium in contaminated sites, and little has been reported on the effects of chromium by colloids and acid rain. A chromite salt production plant in China was chosen as the study area. We studied the effect of acid rain and colloid types on chromium in two soils, humic acid (HA) colloid + cinnamon soil and montmorillonite (Mt) colloid+ cinnamon soil. This study’s findings will help us better understand Cr release in the vadose zone and the possible effects of sulfuric acid rain entering the vadose zone. Understanding the release of pollutants in the vadose zone during acid rain events will support groundwater pollution control measures.

## 2. Materials and Methods

### 2.1. Study Area

The study area is north of the Yellow River alluvial plain and south of the southern foot of Taihang Mountain in Henan Province’s northern region (Figure 1a). The study area experiences frequent and severe winds in the winter and spring due to the continental monsoon. About 650 mm of precipitation falls each year, primarily in showers concentrated in the summer (Figure 1b). The groundwater flows from the northwest to the southeast with a small hydraulic gradient and a slow flow rate because it is high in the northwest and low in the southeast. The study area’s surface soil is cinnamon soil since it is mostly agriculture (Figure 1c), and the bottom soil is silt because it is close to the aquifer (Figure 1d). The study area is situated in central China’s acid rain distribution area’s northern edge, which is a typical sulfuric acid rain region and connected to the use of sulfur coal in northern China [27]. The previous Cr slag stacking spot is located in the north of the research region, where the concentration of Cr reaches 1000 mg/kg and declines from north to south. The silt soil layer and the cinnamon soil layer were both unevenly distributed. On the site, Cr is present in an average quantity of roughly 100 mg/kg. There is a considerable risk of groundwater pollution because of how close it is to groundwater. However, the site still has to add to its quantitative assessment of groundwater pollution.

### 2.2. Experimental Medium

In this study, cinnamon soil (cinnamon soils, also known as brown forest soils, are brown soils formed by the weak leaching and aggregation of carbonates in semi-humid warm temperate regions and have secondary adhesion) and silt soil were collected from the study area and used as the experimental media. The soil samples were air-dried, crushed, and sieved at room temperature, and the particle size distribution was tested with a particle analyzer (Bettersize 2000, Dandong, China). The porosity was assessed by the weighing method. The specific surface area of the soil particles was considered via nitrogen adsorption using a specific surface area analyzer (ASAP 2020, Micromeritics, Norcross, GA, USA). The total organic carbon (TOC) concentration was measured by potassium dichromate titration (TOC-LCPH, Shimadzu, Kyoto, Japan). The test results are shown in Table 1.

The chemical reagents used in this study were of molecular or analytical grade and purchased from Sinopharm Group or Aladdin (Shanghai, China) (if not, it will be noted separately). Analytically pure solid potassium dichromate (K_2_Cr_2_O_7_) was used to prepare the Cr (VI) solution (1000 mg/L) to simulate the source of Cr (VI) in the soil. After drying the potassium dichromate salt at 105 °C to a constant weight, 2.829 g was dissolved in 1 L of Milli-Q water and filtered through a 0.45 µm membrane to obtain a stock solution. The stock solution was diluted for use in the experiments. In the experiments, the diphenylcarbazide method was used to test the Cr (VI) concentration [28,29], the potassium permanganate oxidation method was used to test the total Cr concentration [30], and the Cr (III) concentration was obtained by subtracting. The absorption wavelengths of total chromium and Cr (VI) are 540 nm and 560 nm, respectively.

### 2.3. Batch Experiment

Batch experiments were conducted to identify the influence of acid rain and colloid in the soil on Cr (VI). Humic acid (HA) colloid+ cinnamon soil and montmorillonite (Mt) colloid + silt soil were investigated. Humic acid (HA) colloid+ cinnamon soil: 4 g of humic acid (HA) colloid with a purity of 81.7% (maximum ash content is 10%, maximum moisture content is 8%, maximum iron content is 0.3%, Tianjin institute of national liberation, Tianjin, China) was added to 96 g air-dried ground and sieved cinnamon soil, and thoroughly stirred to make it evenly mixed. Montmorillonite (Mt) colloid + silt soil: 4 g montmorillonite (Mt) colloid (Montmorillonite K-10, purity > 99%, Tianjin Institute of National Liberation, Tianjin, China) was added to the same treated 96 g silt and stirred to make it evenly mixed.

In the experiment, 2 g of the processed, mixed sample was placed in a centrifugal tube, and 10 mL of 5 mg/L contamination liquid was added to the centrifuge tube. The pH was adjusted to 7, 5.5, or 4.5 using 1:1 hydrochloric acid and 8 mol/L sodium hydroxide; these pH values approximately represent neutral, weak, and strong acid rain, respectively. To study the influence of SO_4_^2−^ in acid rain, which is widely present in China, SO_4_^2−^ was also investigated. In the experiment, sulfate concentration was adjusted to the previously set levels of 50 and 100 μmol/L by adding solid potassium sulfate. The Cr (VI) solution was mixed with HA + cinnamon soil and with Mt + silt soil on a shaker at a rotation speed of 150 rpm. Samples of Cr (VI) remaining in the solution were collected over time.

The samples were taken out for testing. Firstly, the concentration of Cr (VI) was assessed by UV-vis spectrophotometry (UV-1800, Shimadzu, Kyoto, Japan) using diphenylcarbazide (Aladdin) as a developer [28]. The 10 mL reaction system was used to test the concentration of Cr (VI). The sample was filtered through a 0.45 μm water filter, and 2 mL filtrate was collected and diluted to 10 mL with water. The filtrate is nearly colorless and clear. A total of 100 µL 1:1 sulfuric acid and 100 µL 1:1 phosphoric acid were added into the reaction system after being shaken well. A total of 400 µL chromogenic agent was added, shaken well, and stood for 5–10 min to test the absorbance and determine the concentration of Cr (VI) according to the standard curve.

Then, all experimental suspensions were tested with a laser nanoparticle size analyzer (SZ-100, Horiba, Kyoto, Japan) after the reaction reached a stable state to determine the binding state of colloid and Cr (VI) in the solution (when testing fluid dynamics diameter, select organic or silicon oxides in the particle preset to test HA colloid and Mt colloid, respectively. The dispersant type was water, the refractive index was 1.333, and the test temperature was set to 25 °C. When zeta potential was tested, the temperature of the particle and dispersant was the same as above, the Henry coefficient term was the Smorukowski formula, and the sample cell used 6 mm graphite electrode cell).

### 2.4. Dynamic Leaching Experiment

A vertical column experimental setup was chosen for the leaching simulation. The length of the plexiglass column was 10 cm, the inner diameter was 2 cm, and the column volume was approximately 31.4 cm^3^. The inner wall of the column was frosted to prevent it from causing preferential flow. Before use, deionized water was repeatedly washed through the column, and microporous filters were installed at both ends of the column to simulate the leaching process and enable even infiltration. A peristaltic pump (BT-100, Longer, Baoding, China) was used to control the water flow at 0.3 mL/min, and an automatic partial collector was used to collect the outflow.

The column was filled with cinnamon soil or silt soil using the wet filling method to exclude air bubbles from interfering with the experiment. In order to better determine the influence of different colloids on chromium adsorption, we set a single variable; that is, when studying the influence of colloid type, the soil column was filled with a single type of soil. The relevant parameters are shown in Table 1. The initial pollution concentration was 100 mg/kg. Water content of approximately 80% was controlled by drainage weighing to simulate the initial unsaturated conditions of precipitation, and the release process of Cr (VI) under different conditions was explored by changing the acid–base, SO_4_^2−^ ion, and precipitation infiltration parameters of the system.

In the continuous infiltration experiment, we conducted 40 h of continuous leaching. The upper part of the leaching medium was cinnamon soil containing the HA colloid, and the lower part was the silt containing Mt colloid. The volume ratio of cinnamon soil to silt is 1:9, which is close to that of the natural vadose zone soil composition. The infiltration solution was 50 μmol/L SO_4_^2−^ at pH 5.5. The intermittent infiltration experiment used pulsed leaching, in which leaching was alternately conducted for 5 h, followed by a pause of 5 h.

### 2.5. Data Analysis

Three replicate analyses were taken for each treatment in the experiment to determine the mean and standard error. All data were analyzed using Origin 2021 and Microsoft Office Excel 2010.

## 3. Results and Discussion

### 3.1. Binding between Cr (VI) and Colloid

From the H1–H3 in Figure 2, it can be found that as the acidity increases, the particle size distribution becomes more dispersed, and the average particle size becomes significantly larger. This was because the acidity changed the structure of the HA colloid. As shown in Figure 2 (H-t), the absolute value of the zeta potential decreased with the increase in acidity. The absolute values of all the resulting zeta potentials of all groups were less than 30 mV. indicating that the systems were in an unstable state [31]. Figure 2 (M1–M3) shows that the average particle size of all groups reached 5000 nm or greater. It can be seen from the figure that as the acidity increases, the particle size distribution becomes more dispersed, but the average particle size changes less. It can be seen that the introduction of hydrogen ions will change the binding state of the particles.

HA colloid will agglomerate under acidic conditions [32], which is caused by protonation [6], and the stronger the acid, the more obvious the agglomeration [33]. Protonated HA colloid with a positive charge on its surface is more conducive to the bonding of negatively charged particles, and its curly structure also reduces its repulsion. Therefore, the particle size in Figure 2 (H1) is more dispersed. At this time, the binding state is more complex, and both large and small particles exist at the same time. However, it can be seen from the zeta potential that the binding is not stable, and there is dynamic binding and release inside. Additional negative charge was introduced as the acidity decreased, which further increased the negative charge on the surface of the Mt colloid. This result was confirmed by monitoring the zeta potential and is consistent with previous research [34]. The particle size increased as the acidity decreased, but this increase was not noticeable, which was different from the findings of a previous study [34]. Introducing a negative charge increases the width of the electric double layer, thereby increasing the particle size. The previous study found the opposite result because the particle was positively charged Cd, while the current study used negatively charged Cr.

Figure 2 (H2,H4,H5,H-t) show that SO_4_^2−^ had a significant effect on particle size. The average particle size is close to 400 nm, indicating that after the introduction of SO_4_^2−^, the influence of protonation on the bonding is significantly weakened. At the same time, it can be seen from the charge change that the increase in electrostatic repulsion also makes the particle size expand, making the bonding more unstable. In the montmorillonite system, from Figure 2 (M2,M4,M5), it can be seen that the particle size shows a gradually increasing trend and is more concentrated in the large particle range. After SO_4_^2−^ was introduced, the electrostatically repulsive double layer was compressed with the increase in ionic strength [35], which significantly decreased the electrostatic repulsion effect and increased the electric potential to approximately −10 mV. In this situation, more particles can overcome the electrostatic repulsion to form larger aggregates. This was confirmed by monitoring the particle size (Figure 2 (M-t)). When the SO_4_^2−^ ion concentration was further increased to 100 μmol/L, the Cr colloid particle size was significantly increased (Figure 2 (M-t)). These results indicate that SO_4_^2−^ plays a considerable role in this process.

In general, acidity will change the structure of the HA colloid and make it more conducive to agglomeration and bonding. The large electrostatic repulsion force of the Mt colloid makes particle agglomeration difficult. However, increasing the ionic strength by introducing SO_4_^2−^ compacts the double electric layer, and the inter-particle van der Waals force may overcome the electrostatic repulsion force, resulting in the formation of larger aggregates [36].

### 3.2. Adsorption Efficiency of Different Colloids for Cr (VI)

In the experimental group without colloid, after 10 h of adsorption, the Cr (VI) fixation efficiency was less than 20% (Figure 2 (H-t)). In the presence of HA colloid, the fixation efficiency was significantly increased. As the acidity increased, the fixation efficiency gradually increased, indicating that acidic conditions promote the effect of the colloid owing to the presence of H^+^ (Figure 2 (H-t)). As the colloid combines with Cr (VI) into large particles, they are deposited onto the surface of the medium. Meanwhile, under acidic conditions, Cr (VI) has oxidation–reduction potential; in this chemical reaction, some of the Cr (VI) is converted to Cr (III) [37,38]. Cr (III) is more able to form complexes and has a relatively low solubility, which promotes its deposition [39]. These characteristics enhance Cr adsorption, and the overall performance is maintained.

Before the introduction of colloids, the adsorption efficiency of Cr in the silt soil was 20.2%. After the introduction of Mt colloid, the adsorption efficiency decreased to 5.2% (Figure 2 (M-t)). In this system, although the aggregate particle size significantly increased from 5200 nm to 5500 nm, the absolute value of the electric potential also significantly increased to more than 40 mV, and greater electrostatic repulsion occurred between the surface of the medium and the solute particles, which leads to adsorption inhibition.

In a HA colloid system, after the introduction of SO_4_^2−^, the original adsorption efficiency of 31.4% dropped to 24.4% (Figure 2 (H-t)). When the SO_4_^2−^ concentration was increased to 100 μmol/L, the adsorption efficiency further decreased to 18.2%, which was lower than the adsorption efficiency without the addition of SO_4_^2−^ (19.7%). This confirms that SO_4_^2−^ inhibits Cr (VI) adsorption, and this inhibition may weaken the effect of HA. After the introduction of SO_4_^2−^, the electrostatic repulsion force of the system is increased, which weakens the adsorption ability of the medium. This is consistent with previous studies of SO_4_^2−^ and HA colloids [40]. Additionally, the protonated HA combines with the negatively charged SO_4_^2−^, reducing the binding ability of HA and further weakening Cr (VI) adsorption.

After the introduction of SO_4_^2−^ in the Mt colloid, the Cr adsorption efficiency slightly increased from 4.4% to 5.1% (Figure 2 (M-t)). We found that the particle size in this system increased further, but the absolute value of the zeta potential significantly decreased. The introduction of SO_4_^2−^ increased the ionic strength of the system, thereby compressing the double electric layer of the Mt colloid and further allowing the Mt colloid to combine with Cr groups to form large aggregates. The large size makes it difficult for Cr to diffuse into the primary potential wells on the surface of the medium. The negative charge of the system further increases, and the repulsion between the medium and the aggregates increases. Therefore, sedimentation is only slightly increased.

In summary, the protonation of the HA colloid, owing to the presence of an acid, allowed Cr to bind the colloid and adsorb to the medium via electrostatic forces. The reduction under acidic conditions promoted the conversion of Cr (VI) to Cr (III), which also led to deposition. However, the negative charge on the Mt colloid surface enhanced the repulsion between the aggregates and the medium, and the adsorption capacity decreased significantly.

### 3.3. Transport Behavior of Cr (VI)-Colloid System in Soil Columns

#### 3.3.1. Effect of HA Colloid on Cr (VI) Release

The HA colloid had an inhibitory effect on the migration of Cr (VI) (Figure 3). In the first hour, the concentration of Cr (VI) dropped from 64.35 mg/L to 32.88 mg/L (Figure 3 (H1,H3)), and the inhibitory effect reached approximately 50%. The main mechanisms by which the HA colloid inhibits Cr (VI) migration are through its ability to reduce Cr (VI) to Cr (III) and through structural changes in the colloid that occur under acidic conditions. Under acidic conditions, protonation changes the colloidal structure of the HA colloid. When the electrostatic repulsion force between particles is reduced to less than the van der Waals force, particle collisions will form larger aggregates. Additionally, the protonation of HA colloid under acidic conditions endows the colloid surface with multiple positive charges, which can electrostatically adsorb multiple Cr_2_O_7_^2−^ ions simultaneously. The positive charge on the surface of the protonated HA colloid will enhance the interaction between itself and the medium and enhances adsorption. HA colloid, which is rich in reducing groups, can react with strongly oxidizing Cr (VI) under acidic conditions to generate Cr (III). In the presence of humic acid, some of the adsorbed Cr (VI) is reduced to Cr (III), thus reducing the toxicity and mobility of Cr (VI) [41]. In the original test group, the Cr (III) concentration was only approximately 1 mg/L (Figure 3 (H1)). After the addition of HA colloid, the concentration of Cr (III) slightly increased (Figure 3 (H3)) because some Cr (VI) was transformed into Cr (III). As mentioned above, Cr (III) can form complexes and has relatively low solubility, promoting deposition [39]. In addition, Cr (III) often exists in the form of a Cr hydroxide precipitate, especially under neutral conditions. Therefore, the concentration of Cr (III) in the outflow did not significantly increase, but it was observed that the outflow concentration of Cr (III) in all groups was increased compared with the condition without HA. By integrating the leaching curve, we found that the leaching efficiency of Cr (VI) over 15 h decreased by 63.88% after adding 2% HA colloid, which indicated that both the release rate and the release efficiency were significantly inhibited.

As the acidity increased, the influence of HA on the migration of Cr (VI) gradually increased, but the impact was small (approximately 5 mg/L). This increase was caused by the acid increasing the protonation, which will change the electrostatic repulsion force [42]. When the electrostatic repulsion force between particles is reduced to less than the van der Waals force, particle collisions will form larger aggregates [43]. Additionally, the protonation of HA colloid under acidic conditions endows the colloid surface with multiple positive charges, which can electrostatically adsorb multiple Cr_2_O_7_^2−^ ions simultaneously. The positive charge on the surface of the protonated HA colloid will enhance the interaction between itself and the medium and enhances adsorption. As mentioned above, Cr (VI) release was inhibited by 63.88% when HA colloid was present, but the inhibition owing to changes in acidity was only 14.47%. This result confirms that acidity is not a key factor. The investigation of Cr (VI) morphology while separately considering acidity revealed that the outflow concentrations of dissolved Cr (VI) (DCr (VI)) and colloidal Cr (VI) (ECr (VI)) will change under different acidic conditions. As the acidity decreased, the leachate DCr (VI) concentration increased. When the pH was increased from 4.5 to 5.5, the concentration of total Cr (VI) (TCr (VI)) in the first hour of leaching increased by approximately 1 mg/L, but the concentration of DCr (VI) increased by approximately 3.5 mg/L. Thus, in this process, there is a tendency for ECr (VI) to convert to DCr (VI). In the liquid phase, the ECr (VI) first converts to DCr (VI) after protonation is weakened, and it migrates out of the system under the hydrodynamic force [44]. When the pH was increased to 7.0, the concentration of TCr (VI) in the outflow increased by 3.8 mg/L; however, the concentration of DCr (VI) increased by less than 1 mg/L during this process. These results indicate that the conversion of the ECr (VI) to the dissolved state slows when the concentration of OH^−^ continues to increase. Meanwhile, the ECr (VI) adsorbed at the solid–liquid interface begins to desorb into the liquid phase owing to the presence of a negative charge, thereby increasing the ECr (VI) concentration in the outflow. This inference was confirmed by subsequent particle size monitoring. It was also confirmed by the leaching efficiency investigation in which the ECr (VI) concentration increased by 25.75% after 15 h compared with the pH 5.5 system.

We monitored the dynamic particle size for two experimental conditions at both pH 4.5 and pH 5.5 and found similar change trends. At pH 7, only one experimental condition was monitored, and the two peaks were close and could not be effectively distinguished (Figure 3 (H2–H4)). We compared the results of the batch experiment and the leaching experiment. We found that the particle sizes that migrated out in the leaching experiment differed from those of the batch experiment. The outflow particle size in the pH 4.5 group was smaller than that in the batch experiment, while those of the other two groups were larger than the particle size of the batch experiment. These results confirm that the morphology changes and protonation of HA under acidic conditions better enable the formation of large agglomerates that adsorb onto the surface of the medium, and these cannot easily migrate out. Most of the particles that can migrate out of the system fail to adsorb to the medium surface. Thus, the particle size will gradually decrease. In the other two groups, as the acidity decreased, the particle size in the outflow significantly increased compared with that of the batch experiment, and the outflow particle size showed an increasing trend. This was mainly because of the accumulation and agglomeration of HA colloids during hydrodynamic collisions. However, the proton concentration on the agglomerate surface is reduced after the protonation is weakened, which makes it difficult for the agglomerates to adsorb to the medium surface through electrostatic interactions. Because they cannot adsorb to the medium surface, the large agglomerates will be released because of the water pressure, and eventually, larger agglomerates will be seen. As shown in Figure 3 (H4), significant periodic changes and increases in particle size were detected. This confirms that the particle size increase is caused by the desorption of colloidal particles from the surface of the medium after the addition of OH^−^.

#### 3.3.2. Effect of Mt Colloid on Cr (VI) Release

The Mt colloid did not inhibit the migration of Cr (VI) but promoted it. After 15 h of leaching, the leaching efficiency increased by 2.64% compared with the system without Mt colloid. This is because although the hydrodynamic particle size is significantly increased in the Mt colloid-containing system, the surface charge of the Mt colloid is negative, and the interaction with the dichromate ion is also negative and strongly repulsive. This leads to poor binding ability, and aggregates cannot easily form. This was later confirmed by analyzing the outflow. The TCr (VI) concentration in the outflow was nearly the same for all groups (Figure 4 (M1–M4)). In this system, the particle size of the Mt colloid–Cr complex was larger than that of the HA colloid. The particle size approached 6000 nm, making it difficult for the particles to fall into the cavities on the medium surface. The contact between the particles and the medium mainly occurred in convex areas of the medium. This phenomenon lengthens the distance between the particles and the medium, which weakens the van der Waals force. Thus, only a small number of aggregates can adsorb onto the surface of the medium. In the Mt colloidal system, although the electrostatic repulsion was relatively large and the van der Waals force was decreased, the overall ability to promote migration was not prominent.

When the acidity was decreased, the TCr (VI) concentration in the outflow increased slightly in the first hour of leaching. However, the leaching efficiency was decreased by 3.11% compared with the pH 5.5 system, indicating that the stronger electrostatic repulsion force increased the migration rate but did not increase the leaching efficiency. In this process, the ECr (VI) concentration barely increased, while the DCr (VI) concentration was increased. These results confirm that a small amount of colloidally bonded Cr (VI) that is adsorbed onto the surface of the medium can be slowly released. Most of the Cr still migrates in the liquid phase in a dissolved form.

It was thought that by adding Mt colloid, nearly all of the conditions would be conducive to colloidal Cr migration, but the results showed that this was not the case. This was because the large particle size of the Mt colloid and the combination of the colloid and Cr (VI) caused blockage of the pore throat. However, there were some differences in this effect under acidic conditions. An increased H^+^ concentration leads to more rapid binding between particles in the pore throat, which causes the pore throat to shrink rapidly. When the H^+^ concentration was decreased, the filtering effect decreased significantly, which was also confirmed by the particle size monitoring. At pH 4.5, the particle size decreased rapidly and then gradually increased, while the other two pH groups showed a slight decrease.

In the Mt system, the Cr (VI) content was mainly in the dissolved state, and the content of Cr (VI) in colloidal form was low, which indicated that the Mt colloid could not easily bind with Cr (VI); thus, most Cr (VI) was still in the dissolved state (Figure 4).

In general, the HA colloid had a significant inhibitory effect on the migration of Cr (VI), and in this process, the aggregated colloids preferentially adsorbed onto the surface of the medium, causing Ostwald ripening. At the same time, the reducing functional groups on the surface of the HA colloid can reduce Cr (VI) (which has a strong migration ability) to Cr (III). Cr (III) is then adsorbed onto the surface of the medium. When the pH is between 4.5 and 5.5, protonation converts the ECr (VI) in the liquid phase to DCr (VI); when the pH is between 5.5 and 7.0, the ECr (VI) adsorbed onto the medium surface releases into the liquid phase and migrated out. When Mt colloid was present, the zeta potential of the system dropped to below −40 mV, and a greater electrostatic repulsion force occurred between the medium and the colloidal particles. At the same time, the presence of large complexed particles increased the effective distance between the particles and the medium, and the van der Waals force further decreased. However, the large particle size of the Mt colloid allows it to easily block the pore throat owing to collisions, which affects migration.

### 3.4. Precipitation Alters the Transport Behavior of Cr (VI)

#### 3.4.1. Effect of Acid Rain on the Leaching Release of Cr (VI)

In China, the main type of acid rain is sulfuric acid rain. Thus, we further studied the influence of HA colloid on the migration of Cr (VI) in the presence of SO_4_^2−^ ions (Figure 3 (H3,H5,H6)). The concentration of Cr (VI) in the outflow rose from 32.88 mg/L to 42.35 mg/L over the course of 1 h. When the SO_4_^2−^ ion concentration reached 100 μmol/L, the Cr (VI) concentration rose to 49.35 mg/L. With the increased SO_4_^2−^ ion concentration, the migration ability of Cr (VI) was significantly enhanced, which is inconsistent with previous studies. Previous studies have suggested that an increased ionic strength will compress the electric double-layer structure of the colloid, leaving it in an unstable state and more likely to adsorb to the medium surface, thus inhibiting migration [35]. In this process, the introduction of SO_4_^2−^ ions increases the negative charge of the system, and a more negative charge increases the electrostatic repulsion between the surface of the medium and the Cr, which enhances migration. The migration concentration of ECr (VI) and DCr (VI) increased. SO_4_^2−^ and dichromic acid have a similar structure, and the two compounds will compete with each other for binding to the protonated HA colloid functional groups. This greatly reduces the amount of Cr that HA colloid can bind and ultimately enhances Cr migration. Comparing H3 and H5 in Figure 3 shows that the concentration of DCr (VI) in the outflow was significantly increased, which confirms our speculation. When SO_4_^2−^ binds to HA, it also increases the negative charge of the colloid surface, increasing the repulsion between the colloid and the medium and transforming the relationship between the colloid and the medium from attraction to repulsion. These assumptions are confirmed by the data in Figure 3 (H5,H6). The migration concentration of ECr (VI) in the two groups was significantly increased, and the concentration of ECr (VI) in Figure 3 (H6) was higher than that in the dissolved state. This result further proves that the originally adsorbed colloid gradually desorbed into the liquid phase and that this desorption process was completed. The change in particle size confirmed this, the particle size in Figure 3 (H5,H6) shows a gradually decreasing trend, indicating that colloidal aggregates were being released from the surface of the medium.

In conclusion, both acidity and the SO_4_^2−^ concentration affected the migration of Cr (VI) in the cinnamon soil containing HA. Considering the results of previous studies and the leaching efficiency, it can be concluded that the acid in acid rain has a weak effect on Cr release (14.47%), while SO_4_^2−^ is the dominant factor (39.77%). Therefore, even if the presence of H^+^ in the soil can slightly enhance Cr fixation, a large amount of Cr will still migrate through the vadose zone under the influence of SO_4_^2−^ from acid rain, which has a significant migration-promoting effect.

In the silty soil in the lower vadose zone, the increased ionic strength compresses the electric double-layer structure of the colloid, reducing the repulsion between particles (Li and Xu 2008). This has been confirmed in previous studies that have monitored the zeta potential and particle size. After the introduction of SO_4_^2−^, the absolute value of the zeta potential decreased significantly, and the particle size increased significantly because of the reduction in repulsion. Furthermore, the concentration of Cr (VI) in the outflow underwent a small change when SO_4_^2−^ was introduced, but this can be ignored. Similarly, the maximum leaching efficiency only increased by approximately 11%. The SO_4_^2−^ ions did not have a significant impact on Cr (VI) migration in silt in the presence of Mt colloid. A comparison of the groups showed that the Cr migration concentration after the introduction of SO_4_^2−^ was mainly composed of ECr (VI). This indicates that Cr (VI), which can adsorb to the surface of the medium, desorbs into the liquid phase under the influence of SO_4_^2−^ and migrates out of the vadose zone into the saturation zone. This conclusion was confirmed by the particle size monitoring results. The particle size of the two groups after the introduction of SO_4_^2−^ showed an increasing trend, and the overall particle size of the high-concentration group increased significantly. The particle size increased initially and then decreased, which also indicated that the colloids blocked the pore throat in the silt soil. The promotion of Cr (VI) migration in the presence of sulfuric acid rain would be mainly due to the SO_4_^2−^ ions, and their influence mainly impacts colloidal Cr (VI).

#### 3.4.2. Effect of Infiltration Mode on the Leaching Release of Cr (VI)

In this study, the concentration in the early stage rapidly decreased in both the continuous infiltration and intermittent infiltration experiments (Figure 5). After approximately 6 h, the leaching rate rapidly decreased. The approximate slope indicated that the release effect in the system significantly decreased while the adsorption effect increased. However, the release effect was maintained to some extent under the action of hydraulic power. Both the continuous leaching and the intermittent leaching experiments showed a tailing phenomenon. The tailing process lasted for approximately 30 h until the end of the experiment, accounting for 75% of the whole test cycle. This indicates that pollutants would be released for a long time after rainfall.

Unlike the continuous leaching experiment, the intermittent leaching process showed periodic fluctuations (Figure 5b), which had some particular patterns. We found that the fluctuation occurred after the leaching was stopped and restarted, which suggests that the fluctuation was related to stopping the leaching. Further study led to the speculation that this sudden increase in concentration was due to the oxidation of Cr (III). After the leaching is stopped, the water in the soil will be discharged out of the system through gravity release. During this time, gas fills the gaps in the upper part of the column, and the presence of the gas–water interface allows oxygen to obtain electrons. The Cr (III) that has been reduced is oxidized back into Cr (VI) by oxygen, which enhances Cr migration. This process may also be related to the presence of manganese [45]. This is often referred to as the phenomenon of yellow return [21]. When a contaminated site receives intermittent precipitation, rapid mutual conversion between Cr (III) and Cr (VI) will occur in the upper cinnamon soil containing a large amount of HA colloid. Unsaturated conditions in the upper soil provide the conditions needed for the conversion of Cr (III) to Cr (VI). By integrating the fluctuation section of the leaching curve, we found that yellow return caused a 0.3563 mg increase in Cr; of this, ECr (VI) contributed 52%, and DCr (VI) contributed 48%, which were nearly equivalent.

By approximating the integration of the leaching curve, we estimated the leaching efficiency of the two precipitation modes. By adding 2% colloid, the leaching efficiency of the silt group increased by 2.64%, and that of the cinnamon soil group decreased by 63.88%. These results show that the colloid type is the main factor affecting the leaching efficiency of Cr (VI). After the introduction of SO_4_^2−^, the leaching efficiency of the silt group increased by 11.98%, but this was not a significant change. By contrast, the cinnamon soil group’s leaching efficiency increased significantly, with a maximum increase of 39.77% and a trend showing a further increase. The influence of SO_4_^2−^ on the Cr (VI) leaching efficiency mainly occurred in the cinnamon soil layer. Accordingly, we speculate that this effect mainly acted on ECr (VI).

By integrating the leaching curve under different simulated rainfall patterns, we found that 4.73 mg/L of Cr (VI) would be released into the aquifer when intermittent precipitation is sustained for 40 h. After continuous precipitation for 40 h, 3.47 mg/L of Cr (VI) would be released into the aquifer. According to the requirement of China Groundwater Quality Standard (GB/T 14848-2017) for Class III water with Cr lower than 0.05 mg/L, pure water with 95 and 70 times the infiltration volume can meet the standard. By assessing the site conditions, we conservatively assumed that the length of groundwater inflow through the study area is 100 m, the average thickness of the aquifer is 15 m, the porosity is 0.4, the thickness of the vadose zone is 10 m, and the average velocity of groundwater after a 40 h intermittent acid rain event would be 20 m/d. Under these conditions, the Cr concentration in the study area downstream reaches 4.30 mg/L. The Cr concentration will reach 3.15 mg/L during continuous acid rain precipitation. These values exceed the standard by 86 times and 63 times, respectively. Hence, there is a large risk of Cr release during acid rain, resulting in severe harm.

## 4. Conclusions

In Cr-polluted areas, abundant organic and inorganic colloids will combine with Cr to form colloidal Cr, which alters Cr adsorption and migration. The frequent occurrence of acid rain in recent years has complicated the transport of pollutants in Cr-contaminated sites. The main conclusions obtained from this study are as follows:HA colloids and Mt colloids in the vadose zone will combine with Cr. The agglomerates formed by HA colloids are more likely to be adsorbed and complexed onto the soil surface. Mt colloids have less influence on Cr. The fixation of Cr by colloid mainly occurs in the cinnamon soil layer containing HA colloid;The adsorption efficiency of Cr was increased by 12.8% with the addition of HA. In the HA-Cr system, the introduction of SO_4_^2−^ inhibited the adsorption of Cr, reducing the adsorption efficiency from 31.4% to 24.4%. The addition of Mt reduced the adsorption efficiency of Cr by 15%. In the Mt-Cr system, the introduction of SO_4_^2−^ had a promoting effect on Cr adsorption, with the adsorption efficiency increasing from 4.4% to 5.1%;Cr release was inhibited by 63.88% when HA colloid was present, but the inhibition owing to changes in acidity was only 14.47%. Mt colloid promotes Cr transport and increases the leaching rate by 2.64% compared to the absence of Mt. However, the effect of acidity change was not significant;Intermittent acid rain will enhance the effect of acid rain. Regarding site risk management, particular attention should be given to monitoring when acid rain stops and then starts again and to reducing water harvesting during this time;The type of colloid considerably impacts Cr release. During conditions of consistent acid rain intensity, the concentration of HA colloid at the site should be increased by applying organic fertilizers to reduce the risk of Cr release in the vadose zone.

## Figures and Tables

**Figure 1 ijerph-19-16414-f001:**
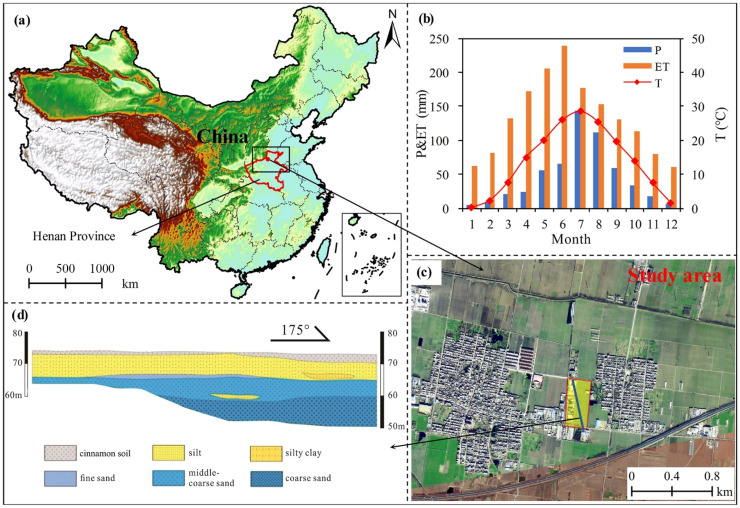
(**a**) Location of the study area. (**b**) Annual precipitation, evaporation and temperature in the study area (P: precipitation; ET: evaporation; T: temperature). (**c**) Enlarged map of the study area. (**d**) Geological conditions in the study area.

**Figure 2 ijerph-19-16414-f002:**
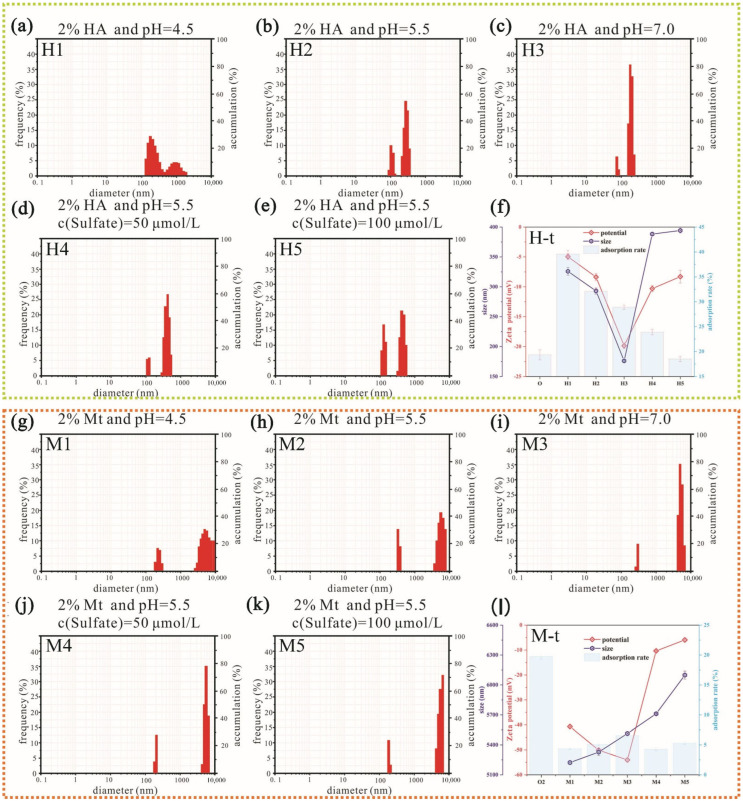
The particle size and zeta potential distribution of the colloid and the adsorption of Cr (VI) by the medium under different conditions. (**a**–**c**) H1–H3 represent the particle size distribution of the HA–Cr system at different pH. (**d**,**e**) H4–H5 represents the particle size distribution of the HA–Cr system at different sulfate concentrations. The O group in ((**f**), H-t) represents the cinnamon soil without colloid; (**g**–**i**) M1–M3 represent the particle size distribution of the Mt–Cr system at different pH. (**j**,**k**) M4–M5 represents the particle size distribution of the Mt–Cr system at different sulfate concentrations. The O2 group in ((**l**) M-t) represents the silt soil without colloid, and the other groups correspond to those in the figure (“2%” is colloidal mass 2% of the total soil mass).

**Figure 3 ijerph-19-16414-f003:**
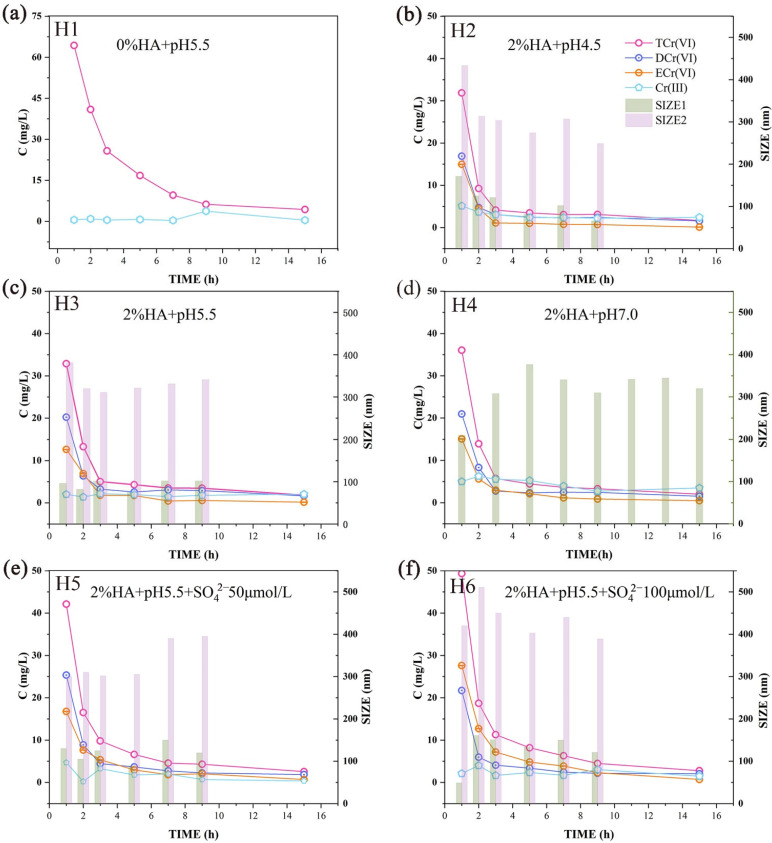
Concentrations of total Cr (VI) (TCr (VI)), dissolved Cr (VI) (DCr (VI)), colloidal Cr (VI) (ECr (VI)), and Cr (III) and the change in HA–Cr system colloid particle size (SIZE1 (the first peak detected by the instrument), SIZE2 (the second peak detected by the instrument) (if available)) during the leaching process. (**a**–**d**) H1–H4 represent concentration of Cr and the particle size of the HA–Cr system at different pH during the leaching process. (**e**,**f**) H5–H6 represents concentration of Cr and the particle size of the HA–Cr system at different sulfate concentrations during the leaching process.

**Figure 4 ijerph-19-16414-f004:**
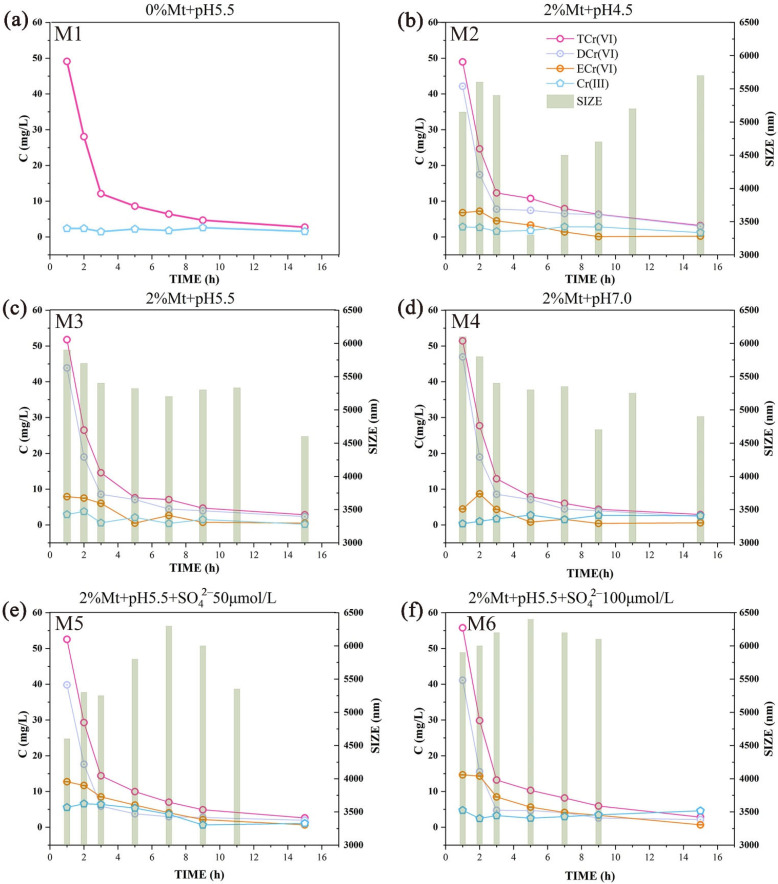
Total Cr (VI) (TCr (VI)), dissolved Cr (VI) (DCr (VI)), colloidal Cr (VI) (ECr (VI)), Cr (III), and the change in Mt–Cr system colloidal particle size (SIZE) during the leaching process. (**a**–**d**) M1–M4 represent concentration of Cr and the particle size of the Mt–Cr system at different pH during the leaching process. (**e**,**f**) M5–M6 represents concentration of Cr and the particle size of the Mt–Cr system at different sulfate concentrations during the leaching process.

**Figure 5 ijerph-19-16414-f005:**
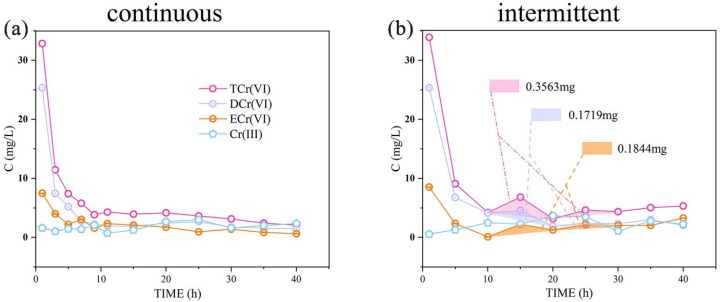
Influence of different precipitation modes on Cr transfer and transformation. (**a**) Continuous represents a continuous leaching experiment of 40 h. (**b**) Intermittent represents a pulsed leaching experiment, in which leaching was alternately conducted for 5 h, followed by a pause of 5 h and the total duration was 40 h. The shaded area indicates the amount of Cr leached during the experiment duration of 10–20 h.

**Table 1 ijerph-19-16414-t001:** Relevant parameters of the research medium.

Type of Medium	pH	Sample Depth (m)	Minimum Particle Size (mm)	Maximum Particle Size (mm)	Specific Surface Area (m^2^/g)	Porosity	TOC (mg/g)
Cinnamon soil							
Mean + standard error	7.77 + 1.12	0.2–0.5	0.21 + 0.01	0.41 + 0.01	0.37 + 0.02	0.432 + 0.010	3.314 + 0.020
Silt							
Mean + standard error	7.82 + 0.92	3.0–4.0	0.19 + 0.00	0.39 + 0.01	0.24 + 0.01	0.337 + 0.008	2.883 + 0.077

## Data Availability

All the data used for the study appear in the article.

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
