# Peer review of "Chromium Transport and Fate in Vadose Zone: Effects of Simulated Acid Rain and Colloidal Types"

_ijerph, 2022, doi:10.3390/ijerph192416414_

Round 1
Reviewer 1 Report
Authors present the article Leaching mechanisms of chromium in soils: effects of acid rain and colloidal types.
Overall, the topic of the article is modern, and the issue is approached systematically. the results are relevant and contribute to the general knowledge related to the topic of heavy metals in the soil and vadose zone.
However, I do have some remarks.
Authors should match the method of citation with that specified in the template. Also, authors should add more newer references regarding the topic of interest.
line 61. please, specify „yellow return“
In the 2.1.study area chapter, figure 2b is mentioned before figure 1. Figures should be mentioned according to their numeration.
Line 93. What is cinnamon soil?
In table 1. Please specify what is standard?
Due to sentence construction, text is difficult to read and understand. Authors need to perform an extensive English grammar and spelling check. Also, in methods section (2.3. batch ex.) should be divided into subsections or at least paragraphs. It would be easier to follow the course of the experiment that way.
The Results chapter should put more emphasis on the results themselves, and the parts concerning the theoretical basis should be moved to the Introduction. In this way, it would be easier to read and understand, because the results are now lost in a large amount of text and information.
Author Response
Response to reviewers
Reviewer 1:
- Authors should match the method of citation with that specified in the template. Also, authors should add more newer references regarding the topic of interest.
Response: Thank you very much for your suggestion. We have changed the style of the references to that in the template and have added some new references regarding the topic of interest.
- line 61. please, specify “yellow return”
Response: Thank you very much for your suggestion. “yellow return”phenomenon is a rebound increase in Cr(VI) concentration in the reduced chromium slag with time in the stockpile.
- In the 2.1.study area chapter, figure 2b is mentioned before figure 1. Figures should be mentioned according to their numeration.
Response: Thank you very much for your suggestion. In the chapter on study areas, we have modified the numbering of Figure 2b.
- Line 93. What is cinnamon soil?
Response: Thank you very much for your suggestion. Cinnamon soils, also known as brown forest soils, are brown soils formed by the weak leaching and aggregation of carbonates in semi-humid warm temperate regions and have secondary adhesion.
- In table 1. Please specify what is standard?
Response: standard is the standard error, which we have modified in Table 1.
- Due to sentence construction, text is difficult to read and understand. Authors need to perform an extensive English grammar and spelling check. Also, in methods section (2.3. batch ex.) should be divided into subsections or at least paragraphs. It would be easier to follow the course of the experiment that way.
Response: Thank you very much for your suggestion.We have already performwed an extensive English grammar and spelling check and in the methods section we have divided it into four paragraphs.
- The Results chapter should put more emphasis on the results themselves, and the parts concerning the theoretical basis should be moved to the Introduction. In this way, it would be easier to read and understand, because the results are now lost in a large amount of text and information.
Response: Thank you very much for your suggestion. We have revised the results to show more conclusive content.
Reviewer 2 Report
The manuscript entitled "Leaching mechanisms of chromium in soils: effects of acid rain and colloidal types” is worth seeking initiatives. This study is an interesting one. It is a relevant to the scope and could be published in the Journal, after some major revisions. Authors need to improve the English language of the manuscript prior to the submission of revision.
Title: Authors are advised to revise the title of study.
Abstract is not clear revise it. First line of the abstract is not convincing. The abstract does not communicate the importance of the work presented.
L 19: Revise it.
There are grammar mistakes in the MS. Professional English Proof reading is mandatory.
Arrange keyword alphabetically.
Introduction:
Authors need to explain and emphasize the novelty of this work in the introduction with comprehensive literature review on the topic.
Revise these lines: L 29, 34, 48, 51, 66, 69.
L31: Give introduction of Cr oxidation states in soil. Then move to Cr (VI).
Objectives and hypothesis of the study are not interesting. Authors should revise.
Methodology should be revised.
Give more detail about diphenylcarbazide method which was used to test the Cr(VI) concentration.
L119: What is the use of potassium dichromate (K2Cr2O7) stock solution.
Revise the L 129, 131, 179, 180
What is meaning of following statement. L 166. Before use, deionized water was repeatedly washed through the column, and microporous filters were installed at both ends of the column to simulate the leaching process and enable even infiltration.
What is wet filling method?
NO detail has been given about Cr (III) and Cr (VI) determination.
Give detail about statistical analysis.
Discussion is not impressive. Support your finding with more references.
Conclusion should be improved. The research conclusion is mainly the research result, and the content of the conclusion is not specific. Authors should add some suggestions for authorities involved in control of pollution.
Revise the L 536.
Captions of the figures should be improved.
Avoid repetition in the illustrations and data.
Arrange the references according to Journal guidelines.
Author Response
Response to reviewers
Reviewer 2:
- The manuscript entitled "Leaching mechanisms of chromium in soils: effects of acid rain and colloidal types” is worth seeking initiatives. This study is an interesting one. It is a relevant to the scope and could be published in the Journal, after some major revisions. Authors need to improve the English language of the manuscript prior to the submission of revision.
Response: Thank you very much for your suggestion. We have revised the article.
- Title: Authors are advised to revise the title of study.
Response: Thank you very much for your suggestion. We have revised the title of study and entitled“Chromium transport and fate in vadose zone:effects of simulated acid rain and colloidal types”
- Abstract is not clear revise it. First line of the abstract is not convincing. The abstract does not communicate the importance of the work presented.
Response: Thank you very much for your suggestion. We have revised the content of the abstract.
- L 19: Revise it.
Response: Thank you very much for your suggestion. We have revised the content of the abstract.
- There are grammar mistakes in the MS. Professional English Proof reading is mandatory.
Response: Thank you very much for your suggestion. We have revised the grammatical mistakes in the study area.
- Arrange keyword alphabetically.
Response: Thank you very much for your suggestion.We have rearranged the order of the keywords.
- Introduction:
Authors need to explain and emphasize the novelty of this work in the introduction with comprehensive literature review on the topic.
Response: Thank you very much for your suggestion. We have explained and emphasized the novelty of this work in the introduction with comprehensive literature review on the topic.
- Revise these lines: L 29, 34, 48, 51, 66, 69.
Response: Thank you very much for your suggestion. We have revised these contents.
- L31: Give introduction of Cr oxidation states in soil. Then move to Cr (VI).
Response: Thank you very much for your suggestion. We have revised the content of introduction.
- Objectives and hypothesis of the study are not interesting. Authors should revise.
Response: Thank you very much for your suggestion. We have revised the objectives and hypothesis of the study.
- Methodology should be revised
Response: Thank you very much for your suggestion. We have revised the methodology.
- Give more detail about diphenylcarbazide method which was used to test the Cr(VI) concentration.
Response: Thank you very much for your suggestion. The procedure for the determination of Cr(VI) is cumbersome and we only introduce the literature on the determination method. Method for the determination of Cr(VI) :Potassium dichromate (K2Cr2O7 analytical grade, Sigma Aldrich) was used to prepare a 1 g/L stock solution of Cr(VI). The DPC method for the determination of Cr(VI) was carried out according to the guideline of the NF T90-043 standard (France). The DPC solution was prepared by mixing 0.02 g of 1,5-diphenylurea with 10 mL of ethanol and 40 mL of 1.8 M sulphuric acid. Complete dissolution of the reagent was achieved after 2 days without heating. For measurement, 1.2 mL of DPC solution and 0.1 mL of concentrated nitric acid were added to 20 mL of sample. Absorption was measured at 560 nm. Linearity between absorbance and concentration was verified over a concentration range of 0-0.8 mg/L Cr(VI).
- L119: What is the use of potassium dichromate (K2Cr2O7) stock solution.
Response: Thank you very much for your suggestion. We prepared a potassium dichromate(K2Cr2O7) stock solution as a provider of Cr(VI) to simulate the source of hexavalent chromium in soil. We have made changes in the article.
- Revise the L 129, 131, 179, 180
Response: Thank you very much for your suggestion. We have revised these contents.
- What is meaning of following statement. L 166. Before use, deionized water was repeatedly washed through the column, and microporous filters were installed at both ends of the column to simulate the leaching process and enable even infiltration.
Response: Thank you very much for your suggestion. Repeated washing of the inside of the column with deionised water is to maintain a single variable environment inside the column. Microporous filters are installed at both ends of the column to evenly distribute water and to prevent the inlet and outlet from being blocked due to media leakage inside the column.
- What is wet filling method?
Response: Thank you very much for your suggestion.Wet filling is the pre-mixing of the soil with deionised water in order to remove air bubbles from the soil that could interfere with the transport experiment. We have added further explanations in the article.
- NO detail has been given about Cr (III) and Cr (VI) determination.
Response: Thank you very much for your suggestion. Total chromium is determined by oxidizing trivalent chromium in water to hexavalent chromium with permanganate, decomposing excess potassium permanganate with sodium nitrite and excess sodium nitrite with urea, and then adding diphenylcarbony dihydrazide for colour development and spectrophotometric determination at 540nm.
The concentration of trivalent chromium is the concentration of total chromium minus the concentration of hexavalent chromium. We have introduced references in the article.
- Give detail about statistical analysis.
Response: Thank you very much for your suggestion.We have given detail about statistical analysis.
- Discussion is not impressive. Support your finding with more references.
Response: Thank you very much for your suggestion. We have added more supporting references to the discussion.
- Conclusion should be improved. The research conclusion is mainly the research result, and the content of the conclusion is not specific. Authors should add some suggestions for authorities involved in control of pollution.
Response: Thank you very much for your suggestion.we have revised the conclusions.
- Revise the L 536.
Response: Thank you very much for your suggestion.we have revised the contents.
- Captions of the figures should be improved.
Response: Thank you very much for your suggestion.We have revised the captions of the figures.
- Avoid repetition in the illustrations and data.
Response: Thank you very much for your suggestion. We have checked the illustrations and data.
- Arrange the references according to Journal guidelines.
Response: Thank you very much for your suggestion. We have arranged the references according to Journal guidelines.
Round 2
Reviewer 1 Report
I would like to thank the authors for the comments and improvements. However, I do have some finishing remarks.
Abstract needs to be rewritten and improved, some sentences sound really awkward.
Figure 1a is not mentioned in the text.
Pls, add definition of cinnamon soil into the text not just in the review reply.
Title 3.2. makes no sense in the new form.
English spelling and grammar still needs improvement.
Author Response
Reviewer 1:
1.Abstract needs to be rewritten and improved, some sentences sound really awkward.
Response: Thank you very much for your suggestion. We have rewritten the abstract.
- Figure 1a is not mentioned in the text.
Response: Thank you very much for your suggestion. We have added a description of
Figure 1a to the 2.1 study area
- Pls, add definition of cinnamon soil into the text not just in the review reply.
Response: Thank you very much for your suggestion. We have added a definition of cinnamon soils to 2.3 Experimental medium.
- Title 3.2. makes no sense in the new form.
Response: Thank you very much for your suggestion. We have revised title 3.2 to“Adsorption efficiency of different colloids for Cr(VI)”
- English spelling and grammar still needs improvement.
Response: Thank you very much for your suggestion. We have revised the grammar and spelling of the whole article.